# Impact of Leading Edge Roughness in Cavitation Simulations around a Twisted Foil

**Abolfazl Asnaghi** and **Rickard E. Bensow** *

Department of Mechanics and Maritime Sciences, Chalmers University of Technology,
SE-412 96 Gothenburg, Sweden; asnaghi@chalmers.se
* Correspondence: rickard.bensow@chalmers.se; Tel.: +46-(0)31-7721479

**Abstract:** The simulation of fully turbulent, three-dimensional, cavitating flow over Delft twisted foil is conducted by an implicit large eddy simulation (LES) approach in both smooth and tripped conditions, the latter by including leading-edge roughness. The analysis investigates the importance of representing the roughness elements on the flow structures in the cavitation prediction. The results include detailed comparisons of cavitation pattern, vorticity distribution, and force predictions with the experimental measurements. It is noted that the presence of roughness generates very small cavitating vortical structures which interact with the main sheet cavity developing over the foil to later form a cloud cavity. Very similar to the experimental observation, these interactions create a streaky sheet cavity interface which cannot be captured in the smooth condition, influencing both the richness of structures in the detached cloudy cavitation as well as the extent and transport of vapour. It is further found to have a direct impact on the pressure distribution, especially in the mid-chord region where the shed cloud cavity collapses.

**Keywords:** cavitation; simulations; roughness; LES; twisted foil

## 1. Introduction

It is generally accepted that an efficient propeller needs to operate in cavitating conditions while still preventing the negative effects of cavitation including noise, vibrations and erosion, which are the results of high-frequency pressure fluctuations induced by the cavity collapse [1,2]. Consequently, cavitation is considered as the main constraint in the propeller design procedure, which has to be carefully assessed both experimentally and numerically to minimise its nuisance.

Numerical simulation is an appropriate complement to experiments, as it can provide further details of the flow field which may not be accessible via the experimental measurements, due to limitations in measurement techniques [3–5]. The numerical simulation of cavitation, however, involves very complex flow physics modelling, such as mass transfer, compressibility, and the simultaneous presence of various temporal and spatial flow scales, making it a challenging fluid dynamic concept. As a result, further development of cavitation computational fluid dynamics (CFD) tools strongly depends on the new insights and knowledge of these complexities that can be extracted from a detailed investigation of numerical results and a well set-up experimental test.

Twisted foils have been used experimentally as the flow, which by design can resemble some structures of three-dimensional propeller cavitating flows, such as interaction of the cavity with vortical structures, cavity collapses, vibration and erosion risk, with the possibility of being tested in a more controlled condition. The cavitation regime of a twisted foil, however, is highly affected by a possible laminar boundary layer forming on the leading edge. Basically, an attached leading-edge sheet cavity does not develop when the boundary layer is laminar, and remains attached to the surface even if the pressure falls below the saturation pressure. The presence of the laminar flow results in significant

scale effects on both forces and cavitation inception. Therefore, in cavitation experiments conducted at low velocities, a widely used alternative approach is to trip the boundary layer into transition by means of local leading-edge roughness elements [6,7].

In the current study, we focus on the numerical modelling of a tripped twisted foil tested at Delft Technical University. This test case was considered as the benchmark of the Workshop on Cavitation and Propeller Performance at the Second International Symposium on Marine Propulsors [8], and consequently, several numerical studies have been performed on the foil [9–14]. In this twisted foil, the natural transition to turbulence occurs on different locations on the hydrofoil, as the loading and local angle of attack of the hydrofoil changes in spanwise direction. Having leading-edge roughness effectively eliminates the laminar flow along with generation of enough flow nuclei. However, in all of the previous simulations, the fact that the leading edge is tripped is not considered explicitly, but it is expected that the modelling represents a fully turbulent flow. There are some recent studies, however in other applications, where a considerable impact of including roughness elements or pits on cavitation inception is reported [15,16].

In our previous studies, numerical simulations of cavitating flows around this twisted foil having a smooth leading edge were carried out [17,18], where it was found that the cavitation model has to be calibrated to provide a proper mass transfer rate. It was also noted that the employed mesh resolution was enough to capture the sheet cavity and the re-entrant jet flow, but was too coarse to accurately convect the cloud cavity. The necessity of employing an advanced condensation mass transfer model capable of handling a cloud cavity consisting of several collapsing bubbles was also regarded. We also highlighted that some parts of the cavitation pattern discrepancy between the numerical results and experimental observation relates to the presence of very fine vortical flows generated behind the roughness elements and their interaction with the main cavity.

As the ability of the solver and the numerical set-up in predicting this complex cavitation flow is demonstrated, we continue this study based on the previous findings regarding the mesh resolution requirement, numerical set-up, and turbulence modelling. In this cavitating modelling approach, it is assumed that both liquid and vapor phases are iso-thermal and incompressible, and the flow follows the homogenouse mixture assumption. Similar mass transfer models for evaporation and condensation having different calibrated coefficients are employed, where the non-condensable gas impact on the mass transfer is neglected. The impact of the nuclei acceleration, the second time derivative of the bubble radius, is neglected in the mass transfer model, and it is assumed that nuclei are evenly distributed in the flow and have a similar initial radius throughout the simulations. Moreover, the turbulence effect on the mass transfer is believed to be very small.

The current investigation contains the impact of roughness on cavitation pattern, vortical structures development, their interactions and effects on the main cloud cavity and consequently on the pressure field. Vorticity distributions are also presented, and the contribution of cavitation on different terms of the vorticity evolution is investigated. A detailed comparison of different vorticity properties, including the stretching, dilatation and baroclinic terms in the tripped and smooth cases is conducted to analyse how the vorticity develops in a cloud cavitation region. The results indicate that in the smooth foil simulation, the cut-off of the sheet cavity leads to the formation of a single large separated cavity structure rather than cloud cavitation. The collapse of this separated cavity in the downstream region creates a spurious secondary cavity, which then results in incorrect pressure prediction over the foil. The results of the tripped case, however, show an accurate formation of a cloud cavitation interacting with several small flow structures generated from the roughness elements.

## 2. Governing Equations

A single-fluid homogeneous mixture model along with the liquid fraction transport equation having an explicit mass transfer modelling is employed to simulate the cavitating flow. To simplify the modelling, both the liquid and vapor phases are considered incompressible and the distribution of the volume fraction function is employed to define the content of each phase. The density and viscosity of

the mixture flow are calculated based on a linear mixture of the each phase property. This mixture assumption is well in line with the filtering assumption of LES models. Using these simplifications, the filtered conservation equations are,

$$\frac{\partial \rho_m}{\partial t} + \frac{\partial (\rho_m \bar{u}_i)}{\partial x_i} = 0, \tag{1}$$

$$\frac{\partial (\rho_m \bar{u}_i)}{\partial t} + \frac{\partial (\rho_m \bar{u}_i \bar{u}_j)}{\partial x_j} = -\frac{\partial \bar{p}}{\partial x_i} + \frac{\partial}{\partial x_j} (\bar{S}_{ij} - B_{ij}), \tag{2}$$

where the over bar represents the low-pass filtering operator. The gravity is neglected, as it is noted in our previous studies that it has a minimal impact on the cavitation pattern. The strain rate tensor is defined by $\bar{S}_{ij}$, which is proportional to the deformation rate tensor, $D_{ij}$,

$$\bar{S}_{ij} = 2\mu \bar{D}_{ij}, \tag{3}$$

$$\bar{D}_{ij} = \frac{1}{2} \left( \frac{\partial \bar{u}_i}{\partial x_j} + \frac{\partial \bar{u}_j}{\partial x_i} \right). \tag{4}$$

In this study, we use implicit LES, ILES, where no explicit model is employed to calculate the sub-grid stress tensor, $B_{ij} = \rho(\overline{u_i u_j} - \bar{u}_i \bar{u}_j)$; instead, it is considered that the numerical dissipation is in the range of sub-grid turbulence viscosity [19,20].

## 2.1. Phase Change Modelling

As noted above, a transport equation is introduced for the liquid fraction in this multiphase modelling approach, and an explicit source term is employed to represent the mass transfer,

$$\frac{\partial \alpha_1}{\partial t} + \frac{\partial (\alpha_1 \bar{u}_i)}{\partial x_i} = \frac{\dot{m}}{\rho_1}. \tag{5}$$

Furthermore, the density and viscosity of the mixture flow properties are calculated by,

$$\rho_m = \alpha_1 \rho_1 + (1 - \alpha_1)\rho_v, \quad \mu_m = \alpha_1 \mu_1 + (1 - \alpha_1)\mu_v. \tag{6}$$

In our previous studies, the Schnerr–Sauer model, Equation (7), is modified based on the velocity strain rate time scale and the mean flow time scale to improve the mass transfer rate, Equation (8),

$$\dot{m} = C_{\text{v-mod}} \text{sign}(p - p_{\text{threshold}}) \frac{\alpha_1(1 - \alpha_1 + \alpha_{\text{Nuc}})}{R_B} \frac{3\rho_1\rho_v}{\rho_m} \sqrt{\frac{2\,|p - p_{\text{threshold}}|}{3\rho_1}}. \tag{7}$$

$$C_{\text{v-mod}} = \left( 1 + t_\infty \left\| \frac{1}{2} \left( \frac{\partial u_i}{\partial x_j} + \frac{\partial u_j}{\partial x_i} \right) \right\| \right). \tag{8}$$

In this definition, $\alpha_{\text{Nuc}}$ is the initial volume fraction of the nuclei and is defined by $\alpha_{\text{Nuc}} = \frac{\frac{4}{3}\pi n_0 R_{\text{Nuc}}^3}{1 + \frac{4}{3}\pi n_0 R_{\text{Nuc}}^3}$, $R_B$ is the bubble radius, $n_0$ is the average nuclei per liquid volume and $R_{\text{Nuc}}$ is the initial nuclei radius. The time scale of the mean flow, $t_\infty = \frac{L_\infty}{U_\infty}$, is also employed to normalize the velocity strain rate value where $L_\infty$ is the reference length scale, i.e., the foil chord length, and $U_\infty$ is the reference velocity, i.e., the inlet velocity. Further information regarding the concepts and results of this modification can be found in [17]. In these equations, the average nuclei per liquid volume and the initial nuclei diameter are set constant and equal to $n_0 = 10^8$, and $d_{\text{Nuc}} = 10^{-6}$ m.

Very close to the inception or collapse state where $\alpha_1$ is close to one, $R_B$ is close to $R_{\text{Nuc}}$ and $\rho_m$ tends to $\rho_1$, the mass transfer rate can be written as,

$$\dot{m} = C_{\text{v-mod}} \text{sign}(p - p_{\text{threshold}}) \frac{\alpha_{\text{Nuc}}}{R_{\text{Nuc}}} (3\rho_v) \sqrt{\frac{2 |p - p_{\text{threshold}}|}{3\rho_1}}. \tag{9}$$

Then, it can be deduced that,

$$\dot{m} \propto \frac{\alpha_{\text{Nuc}}}{R_{\text{Nuc}}} \sqrt{|p - p_{\text{threshold}}|} = \frac{1}{R_{\text{Nuc}}} \frac{\frac{4}{3}\pi n_0 R_{\text{Nuc}}^3}{1 + \frac{4}{3}\pi n_0 R_{\text{Nuc}}^3} \sqrt{|p - p_{\text{threshold}}|}. \tag{10}$$

This indicates that a low value of $n_0$ or $R_{\text{Nuc}}$ leads to a low mass transfer rate and vice versa. For turbulent cavitating flows, it is observed that evaporation happens as soon as pressure falls below the saturation pressure. This relates to the rich nuclei environment provided by the turbulent flow. Therefore, $n_0$ and $R_{\text{Nuc}}$ parameters should be selected as high as possible without compromising numerical stability, in order to provide near instantaneous evaporation. However, as the non-condensable gas effect is ignored, the condensation term should be selected in a way to allow for some retardation in the condensation. Contradictory to the evaporation, this demands for low values of $n_0$ or $R_{\text{Nuc}}$. Thus, it is proposed by the authors to use small values of these parameters, e.g., $n_0 = 10^8$, and $d_{\text{Nuc}} = 10^{-6}$ m, and then modify the evaporation mass transfer coefficient, Equation (8). For further details, please see [17].

The threshold pressure, $p_{\text{threshold}}$, is the pressure where the mass transfer between phases happen. For a static fluid, this pressure is equal to the saturation pressure. However, as the rupture of a liquid pocket in a flowing fluid depends on both the pressure tensile and the viscous stresses, the threshold pressure should depend on the magnitude and direction of the viscous stresses. In our previous study, Ref. [17], it is discussed that the effect of the shear stresses can be incorporated by considering the magnitude of the shear strain rate tensor,

$$p_{\text{threshold}} = p_{\text{Sat}} + \mu \sqrt{2 D_{ij} D_{ij}}. \tag{11}$$

### 2.2. Vorticity Transport Equation

Vorticity is defined as the curl of the velocity,

$$\omega_i = \epsilon_{ijk} \frac{\partial u_k}{\partial x_j}, \tag{12}$$

where in this equation $\omega_i$, $u_k$, and $\epsilon_{ijk}$ are the vorticity vector, the velocity vector and the alternating tensor. The vorticity transport equation obtained by applying the curl function over the Navier–Stokes equations is,

$$\frac{\partial \omega_i}{\partial t} + u_j \frac{\partial \omega_i}{\partial x_j} = \omega_j \frac{\partial u_i}{\partial x_j} - \omega_i \frac{\partial u_j}{\partial x_j} + \frac{1}{(\rho_m)^2} \epsilon_{ijk} \frac{\partial \rho_m}{\partial x_j} \frac{\partial p}{\partial x_k} + \nu \frac{\partial^2 \omega_i}{\partial x_j \partial x_j}, \tag{13}$$

where the left-hand side (LHS) of the equation is the material derivative of the vorticity. The description of the right-hand side (RHS) terms of this equation is presented in Table 1. In cavitating flows and due to the mass transfer between phases, the velocity divergence is non-zero and the vorticity dilatation is important in the region where mass transfer occurs.

To identify vortical structures, the *Q*-criterion, which represents the balance between the deformation and shear strain rates, is employed [21,22],

$$Q = \frac{1}{2} (\bar{\Omega}_{ij} \bar{\Omega}_{ij} - \bar{D}_{ij} \bar{D}_{ij}), \tag{14}$$

where

$$\bar{\Omega}_{ij} = \frac{1}{2}\left(\frac{\partial \bar{u}_i}{\partial x_j} - \frac{\partial \bar{u}_j}{\partial x_i}\right). \tag{15}$$

The pressure coefficient is represented by $C_P = (p - p_{\text{ref}}) / \left(\frac{1}{2}\rho U_{\text{ref}}^2\right)$ where, in this study, $p_{\text{ref}}$ is the outlet pressure and $U_{\text{ref}}$ is the inlet velocity.

**Table 1.** Description of the vorticity transport equation terms.

| Terms | Description |
|-------|-------------|
| $\omega_j \frac{\partial u_i}{\partial x_j}$ | Vorticity stretching and turning due to the flow velocity gradients. |
| $\omega_i \frac{\partial u_j}{\partial x_j}$ | Vorticity dilatation due to the velocity divergence. |
| $\frac{1}{(\rho_m)^2}\epsilon_{ijk}\frac{\partial \rho_m}{\partial x_j}\frac{\partial p}{\partial x_k}$ | Vorticity baroclinic torque due to misalignment of the density and the pressure gradients. |
| $\nu\frac{\partial^2 \omega_i}{\partial x_j \partial x_j}$ | Vorticity diffusion due to the flow viscosity. |

## 3. Computational Domain

The geometry of the so-called twist11 hydrofoil has a NACA0009 profile, with symmetrical varying angle of attack over its span from zero degree on the sides to 11 degrees at the center of the foil. The current simulations are conducted on the foil with the angle of attack of $-2$ degrees, which therefore gives the angle of attack of 9 degrees at the foil center line. The chord length of the foil is $C = 150$ mm, and its span length is $S = 300$ mm. The spanwise varying angle of attack provides different loads over the foil, which results in three-dimensional sheet cavity having a convex-shaped closure line.

The experimental tests carried out in the cavitation tunnel at Delft Technical University are selected to compare with our numerical results [6,7]. The pressure on the foil surface is measured by using pressure probes placed in different positions according to Figure 1. In this figure, the probe numbering is the same as in the experimental test report. The experimental tests were repeated in the EPFL cavitation tunnel on a geometrically similar foil. Due to the limitation of the cavitation tunnel sizes, the EPFL tested foil had a scale factor 0.5.

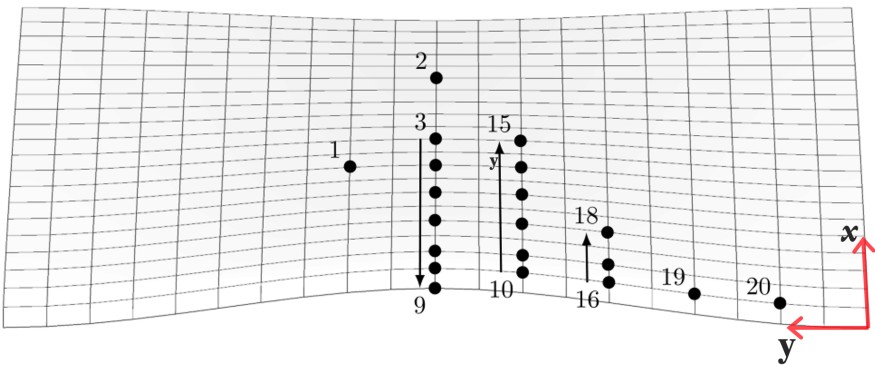

**Figure 1.** Location and numberings of the pressure probes used in the Delft twist11 foil tests [6].

As mentioned, the spanwise varying angle of attack provides spanwise varying laminar to turbulent boundary layer transition line. The presence of a laminar boundary layer often suppresses inception considerably and makes the cavitation pattern more dependent on the nuclei distribution and free-stream turbulence properties. Therefore, to avoid any possible impact of this on the cavitation pattern, the leading edge is roughened by using sand grains to trigger the leading edge boundary layer across the foil span into a fully turbulent boundary layer. The application of applying sand grains provides sufficient nuclei all over the foil span and removes the laminar boundary layer. As a result, the inception will be mostly dependent on the spanwise pressure distribution. In the body of literature

where this flow is simulated, the roughness has not been explicitly treated and it has been implicitly considered that the boundary layer is turbulent and pressure dependant cavitation inception occurs. This, however, neglects the turbulent streaks caused by the roughness elements, which we will see greatly affect the cavity development.

The operating condition is set based on the guideline provided in the workshop of 2nd International Symposium on Marine Propulsors where the uniform fixed inflow boundary is 6.97 m/s, at the inlet and the constant pressure boundary at the outlet is 29,000 Pa. This provides the outlet cavitation number of 1.07. The other boundaries, including the lower, upper, and side walls, are treated as slip boundaries. To reduce the computational cost, only half of the cavitation tunnel is simulated, and then a symmetry plane is used at the center of the tunnel [8,12]. In Figure 2, the computational domain is presented, where sizes are selected to represent the cavitation tunnel of Delft. The Cartesian coordinate is placed at the leading edge of the foil and attached to a side wall where $y$ points towards the foil mid-section and $x$ points to the foil trailing edge.

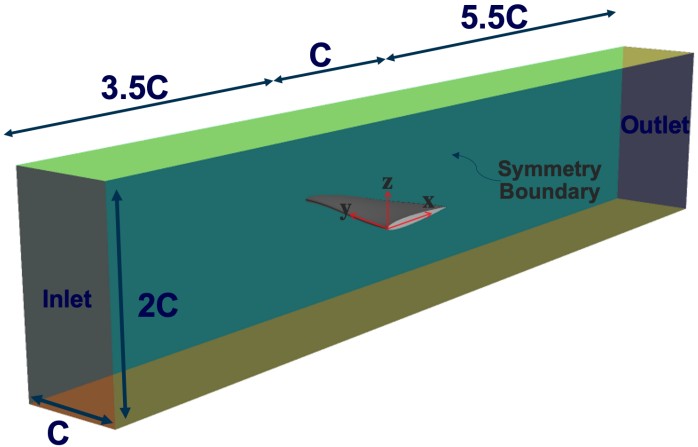

**Figure 2.** Computational domain employed in the current numerical simulations.

In our previous studies [17,18], the computational guideline to successfully model the cavitating flow around this foil having smooth surface within OpenFOAM was investigated. In this guideline, the turbulence modelling impact, minimum required spatial mesh resolution for modelling the cloud cavitation and the numerical set up that can provide low numerical dissipation and high stability are discussed. Here, the same numerical guideline is employed.

The computational domain mostly consists of hexahedral cells with extruded boundary layer prisms generated by StarCCM+. The boundary layer is fully resolved in the wall normal direction, according to our coordinate system $z^+ < 1$ where $z^+ = u_\tau \Delta z / \nu$ and z is the normal distance from the foil surface. The prismatic layers consist of 20 layers having an extrusion factor of 1.05 to provide an appropriate boundary layer resolution around the foil.

Several refinement boxes are employed to specify resolutions around the foil with a specific focus on the cloud cavitation transportation, Figure 3. Two main refinement regions, RR1 and RR2, are specified in the figure. RR1 has been defined based on the flow analysis of our previous studies on the cavitating behaviour around this foil. It emphasizes the region where sheet cavity separates from the foil and forms a cloud cavity on the suction side. RR2 follows the topology of the foil and forces a specific resolution 0.15C distance from the foil surface in both the suction side and pressure side. The normalised resolutions specified by these regions are $RR1^+ < 50$ and $RR2^+ < 100$, where $RR1^+ = u_\tau \Delta RR1 / \nu$ and $RR2^+ = u_\tau \Delta RR2 / \nu$. In these definitions, $\Delta RR1$ and $\Delta RR2$ are the cell resolutions in RR1 and RR2 regions, respectively.

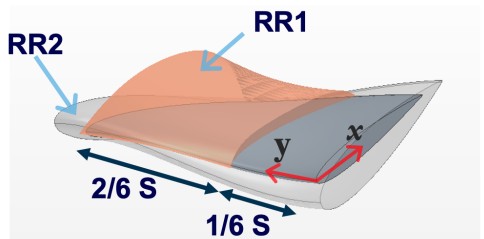 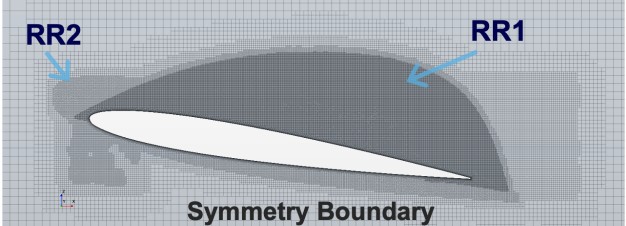

**Figure 3.** Refinement regions (RR) applied to generate computational mesh; *S*: the foil span length.

The surface mesh resolution on the foil gives $(x^+, y^+) < (100, 100)$, with much finer grid resolutions provided on the trailing edge and leading edge of the foil due to the high geometry curvature. Here, $x^+$ and $y^+$ are non-dimensionalised resolutions in x and y directions calculated similar to $z^+$ term, i.e., $x^+ = u_\tau \Delta x / \nu$ and $y^+ = u_\tau \Delta y / \nu$. In these definitions, $\Delta x$ and $\Delta y$ are the cell resolution in x and y directions. In the suction side area of $Y = y/S > 1/6$, where RR1 is applied, the surface resolution is finer, i.e., $(x^+, y^+) < (50, 50)$. In this definition, S is the foil span length and is equal to 0.3 m.

## 4. Leading Edge Roughness

At the considered inlet velocity, i.e., 6.97 m/s, and due to the spanwise varying angle of attack, the natural laminar to turbulence boundary layer transition happens on different spanwise locations on the foil leading edge. Therefore, to avoid the scale effect on the inception of attached cavitation, the laminar boundary layer is forced into transition by using the leading-edge roughness.

In the experimental tests, roughness elements with an average roughness height of 100 µm were applied on the leading edge to cover 4% of the chord length. As is discussed in the experimental test report [6], the presence of roughness and its uneven distribution can cause local streaks of cavitation. It was deduced that cavitation inception on individual roughness elements could be a result of either having locally too large roughness elements or too small roughness elements. In the latter, the flow between roughness elements is insufficiently disturbed and would remains laminar.

In this study, roughness elements are included in the latest step of the mesh generation process by using OpenFOAM utilities. The steps of the required actions are listed below,

- select the surface area where the roughness should be applied, i.e., in the 4% chord length from the leading edge,
- create the face list of all of the faces on the selected surface area,
- modify the face list to adjust the desired roughness elements concentration,
- find and mark all of the cells which are in a certain distance from the modified face list; this distance represents the roughness elements height,
- create a cell list from the marked cells,
- remove the marked cells from the computational domain.

The approach is simple to be applied on any available mesh, and gives the possibility to test different roughness concentrations. However, as it is applied on the original mesh resolution, the computational resolution around each roughness element and its geometry are defined by the initial resolution of that area, Figure 4. This is not an exact representation of the sand roughness used in the experiments, but is expected to give a qualitatively similar effect on the flow.

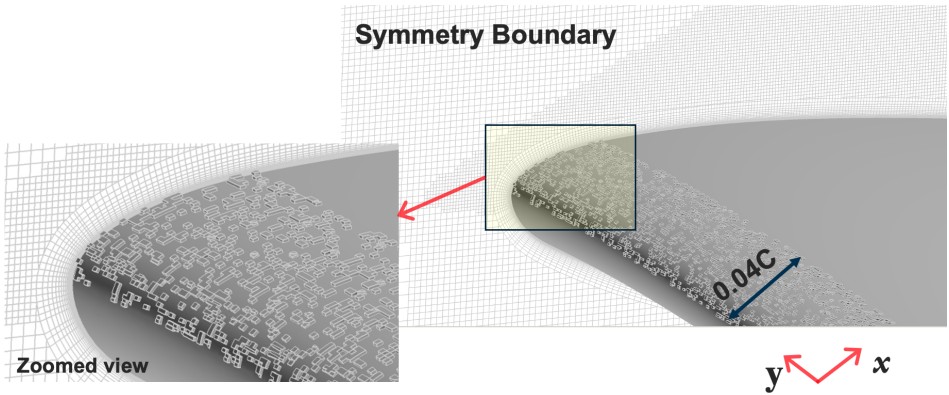

**Figure 4.** Roughness elements distribution in the leading edge of the foil.

## 5. Solution Procedure and Discretization

The governing equations are solved using OpenFOAM, a set of open source community-based CFD libraries [23,24]. A modified version of interPhaseChangeFOAM solver is employed to provide the coupling between the equations using the PIMPLE algorithm. To assure a proper coupling between the transport equation model and the conservation equations, at least two outer SIMPLE loops are used in the PIMPLE set-up, where in each SIMPLE loop, at least three pressure corrections, i.e., PISO loops, are used. When the flow is converged, the convergence of resolving equations is forced by setting a maximum allowed residual in each iteration, i.e., $10^{-8}$ for the wetted flow simulations and $10^{-12}$ for the cavitating flow simulations. These values are set small enough to guarantee that the iterative error is insignificant comparing to the error introduced by the discretization [25].

All of the time discretizations are handled by a second-order implicit scheme along with a small time step set by forcing the maximum Courant number to be less than one. A linear blending scheme is employed for the convective terms, which blends 75% of the second order linear scheme with 25% of the first order upwind scheme. The gradients and interpolations of flow properties on the computational faces are corrected to consider non-orthogonality of the cells. In Table 2, the density and viscosity of the phases, i.e., liquid and vapour, are presented.

It is worth mentioning that the capability of the current numerical set-up and solver in accurately capturing cavitating flows in different applications and flow regimes are validated in our previous works; see [26–30].

**Table 2.** Phases properties.

| Phase | Density (kg/m$^3$) | Dynamic Viscosity (m$^2$/s) |
|---|---|---|
| Liquid | 1000 | $10^{-6}$ |
| Vapor | $2.3 \times 10^{-2}$ | $4.27 \times 10^{-4}$ |

## 6. Results

### 6.1. Pressure Distribution

The distribution of the averaged pressure coefficient of the cavitating condition at two sections of the foil is presented in Figure 5. The experimental measurements conducted at Delft and EPFL cavitation tunnels are included as well. Note that in this figure, $Y$ is the non-dimensioned y-direction location normalized by the span length, i.e., $Y = y/S$, where $Y = 0.5$ represents the symmetry plan at the mid-span location, and $X$ is the non-dimensioned x-direction location normalized by the chord length, i.e., $X = x/C$, where $X = 0$ represents the leading edge and $X = 1$ represents the trailing edge.

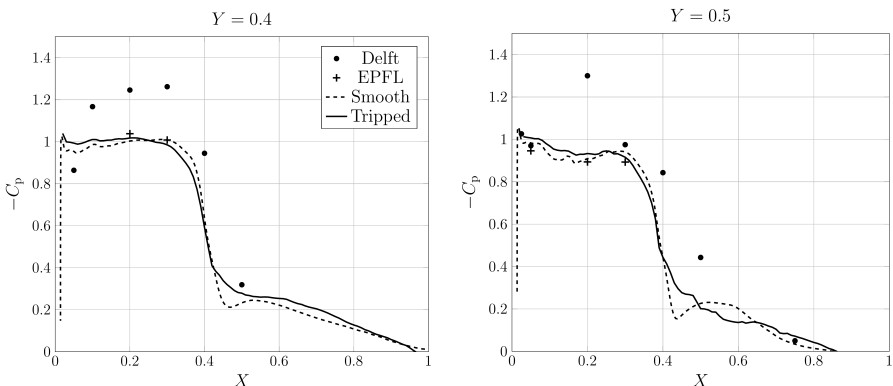

**Figure 5.** Distribution of the pressure coefficient at different sections of the Twist11 foil.

The measured data indicate three different regions: region (I) located in $0 < X < 0.3$, region (II) located in $0.3 < X < 0.6$, and region (III) located in $0.6 < X < 1.0$. In region (I), the measured pressure is close to the saturation pressure, which indicates a dominant sheet cavitation over this region. Region (II) experiences a large variation of pressure due to the separation of the sheet cavity from the foil and consequently the formation of cloud cavitation. Then, the cloud cavity is convected downstream, through region (III), where it experiences higher surrounding pressure and eventually collapse.

The numerical results of the smooth and tripped cases have a similar level of agreement with the experimental measurements in region (I) and the small differences here are not significant and may be due to numerical uncertainty or the level of statistical convergence; the pressure here should be around the saturation pressure, corresponding to $C_P = -1$, when there is an attached cavity that is somewhat higher than the saturation pressure during the shedding process. The main discrepancy between the numerical results and the measurements can be noted in region (II). As mentioned, in this region, the cloud cavity forms, including several structures with bubbles having different sizes and vapour concentration. As the cloud cavity convects downstream, due to the surrounding pressure, some of these bubbles collapse and the cloud shrinks. This leads to a gradual increase of pressure over the foil. In the smooth foil simulation, the sheet cavity is separated as a single main cavity which then forms a single coherent travelling structure, rather than an actual cloud cavity (this will be discussed in detail in the cavitation pattern section). The collapse of this large travelling cavity poses a large pressure variation over the foil; noticeable in $0.45 < X < 0.55$ at $Y = 0.4$ and in $0.425 < X < 0.575$ at $Y = 0.5$. In this regard, the tripped case has better agreement with the experimental measurement, which is believed to be related to the more accurate formation of the cloud cavitation in this case.

*6.2. Lift Coefficient*

The time variation of lift coefficient for the smooth and tripped cases are presented in Figure 6. The time period, T, used in this figure to normalize the time series is computed from the frequency of each lift coefficient dataset.

The main frequency of the flow computed based on the dominant frequency of the lift force variation over time is presented in Table 3, along with the experimentally determined frequency of 32.5 Hz. In both of the simulated conditions, a relatively acceptable agreement between the predicted flow frequency and the measured data is observed, where a closer match is found in the tripped condition. The table also includes the time averaged lift coefficient results, collected over five simulated cycles. As can be seen, the comparative error level of the lift force prediction in the smooth condition is around 11.5%, and in the tripped condition is around 18.2%. Although these comparative errors may seem large, they are actually not larger than the possible uncertainty of the experimental measurements (private communication). In our previous study, it is highlighted that the lift coefficient numerical uncertainty in the employed numerical set-up and mesh resolution is around 22%; please refer to [18] for further discussion.

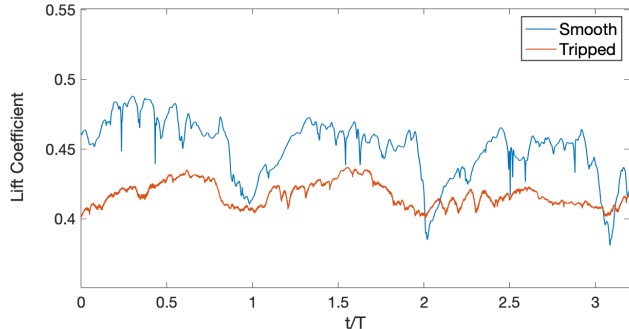

**Figure 6.** Variation of lift coefficient in the cavitating flow conditions.

**Table 3.** Comparison of the shedding frequency and the time averaged Lift coefficient for the smooth and tripped cases.

| Mesh | $C_{\text{lift}}$ | $C_{\text{lift}}$ Comparative Error (%) | $f_{\text{Hz}}$ | $f_{\text{Hz}}$ Comparative Error (%) |
|---|---|---|---|---|
| Smooth | 0.453 | −11.5 | 34.1 | 4.9 |
| Tripped | 0.417 | −18.2 | 33.0 | 1.5 |
| Exp. [6] | 0.51 | - | 32.5 | - |

### 6.3. Cavitation Pattern

The first part of one typical shedding cycle of the cavitation pattern is presented in Figure 7 along with the numerical results of unsteady cavitation shedding behaviour. The figures contain the top view of the foil for smooth and tripped simulations. The iso-surface of vapour and *Q*-criterion coloured by the magnitude of vorticity are provided to elaborate the comparison with the experimental observations. The results are displayed together to easily assess the interaction of vortical structures and vapour distribution.

The general cavitation pattern starts with a sheet cavitation growing from the foil leading edge where its closure can even reach the foil mid-chord. Inside this sheet cavity, a flow circulation zone forms which as a result leads to the formation of liquid jets beneath the sheet cavity in the opposite direction of the inflow; so called the re- and side-entrant jets; followed by the cut-off of the sheet cavitation, Figure 7, shedding of the separated structure as a cloud cavity to the downstream, and finally the collapse of shed cloud cavity.

Due to the geometrical feature of the foil, the re-entrant jet forms a radial flow where it is more concentrated at the centre of the foil, Figure 7a,b. While in the smooth foil results, a sharp uniform interface is observed; the cavity interface of the tripped case clearly shows a wavy shape similar to the experimental observations, due to the impact of the leading edge roughness.

As mentioned, the re-entrant flow moving in the opposite direction of the inflow is the result of a circulation zone formed inside the sheet cavity. When the re-entrant flow reaches the leading edge, it means that the circulation zone contains the whole sheet cavity and the cavity no longer has any connection with the foil surface. Therefore, the inflow cuts the sheet cavity off the foil and creates a large swirling flow around this separated cavity, Figure 7c.

At this state, due to the interactions of vortical structures generated by the leading edge roughness elements with the main circulation of the separated cavity, the cavity quickly splits into several very small turbulent vapour cloud structures. One possible reason is the difference between the strength of circulation zones generated behind the roughness elements in different foil span locations. This difference leads to different spanwise re-entrant flow strength, which consequently results in having different spanwise cut-off over the foil leading edge.

Due to the foil geometry, the separated cavity becomes concentrated around the centre line of the foil as it travels downstream, Figure 7d,e.

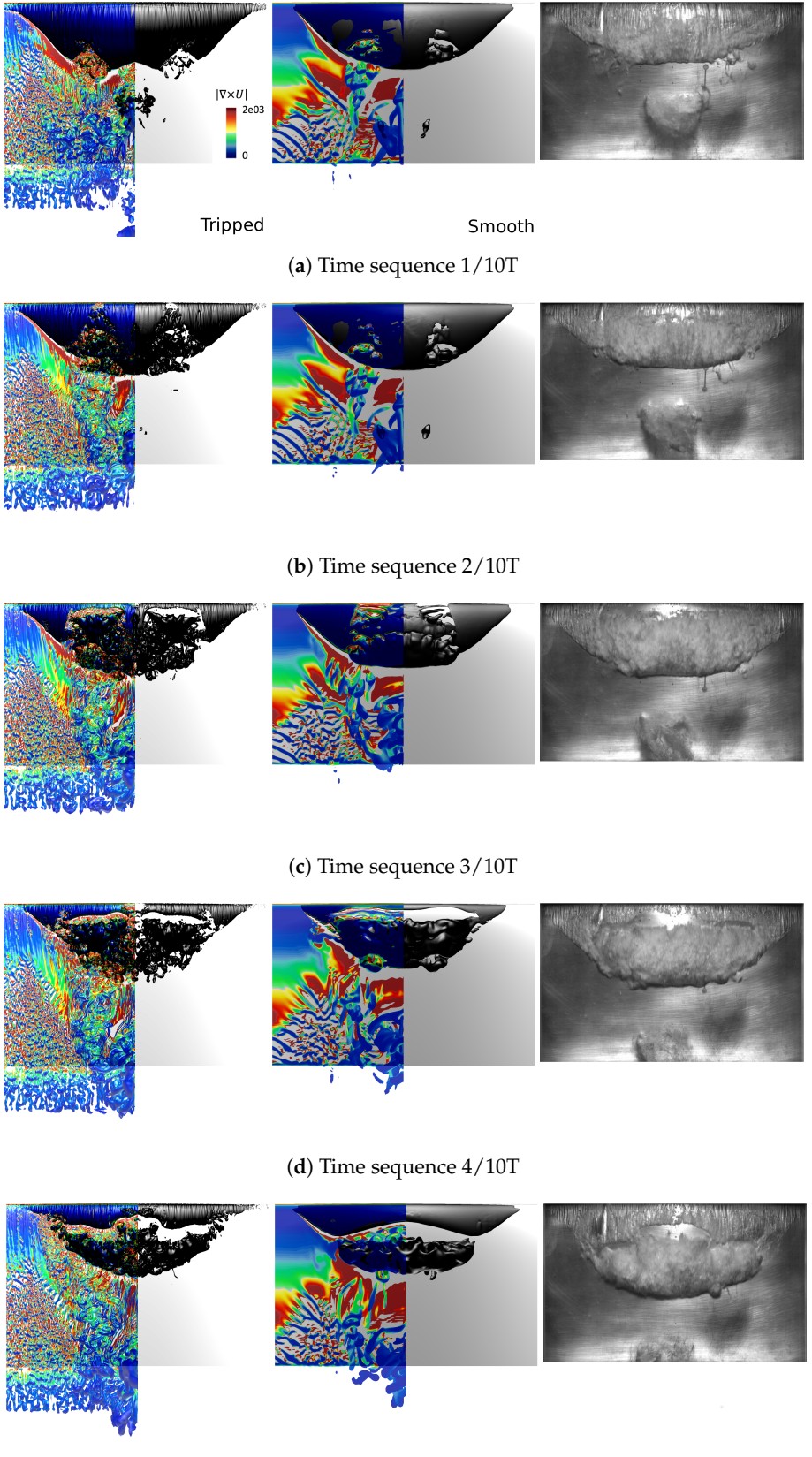

(**a**) Time sequence 1/10T

(**b**) Time sequence 2/10T

(**c**) Time sequence 3/10T

(**d**) Time sequence 4/10T

(**e**) Time sequence 5/10T

**Figure 7.** Sheet and cloud cavitation shedding development.

Getting into more details, an iso-metric view of the results related to the instance Figure 7e is presented in Figure 8. For the smooth foil, the vortical structures are dominated by a few large

structures basically formed around a main separated cavity, while in the tripped case, a cloud cavitation is formed due to the high level of interactions between the vortical structures and sheet cavity. This is in good agreement with the experimental observation indicating the formation of a cloud of bubbles. The predicted shape of the separated cavity in the smooth condition has a direct impact on the foil pressure distribution and the cavitation transportation pattern.

As discussed earlier in Figure 5, the pressure distribution of region (II) located in $0.3 < X < 0.6$ is governed by this cloud cavitation formation. Contradictory to the experimental measurements, in the smooth condition, a decrease of the foil surface pressure is observed in $0.45 < X < 0.55$. This pressure decrease is believed to be due to how cavity shedding is predicted in the smooth condition. Here, the shed cavity forms a rather coherent structure with a smooth surface, almost like a glassy cavity. When this shed cavity is transported downstream and later collapses, it has a large impact on the pressure field. In the tripped case, the cloud cavity consists of several smaller (bubbly) structures which, as the cavities travel downstream, continuously collapse due to the surrounding pressure. However, as their sizes are small, their impact on the surface pressure field is negligible compared to the main cloud cavity pressure field, then leading to a more gradual increase in the pressure over the foil surface. This finding highlights the necessity of including the roughness (or an alternative approach) to model the cloud cavitation. This is of importance for erosion assessment, where the stored energy of cavitation structures is assumed to determine the erosiveness of a cavitating flow [31,32].

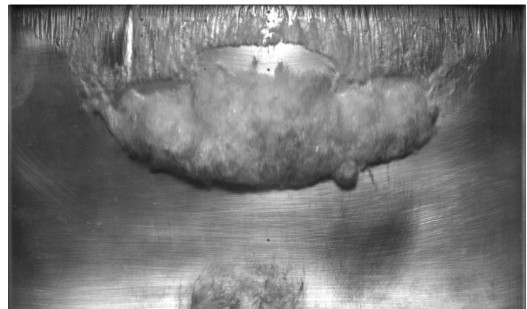

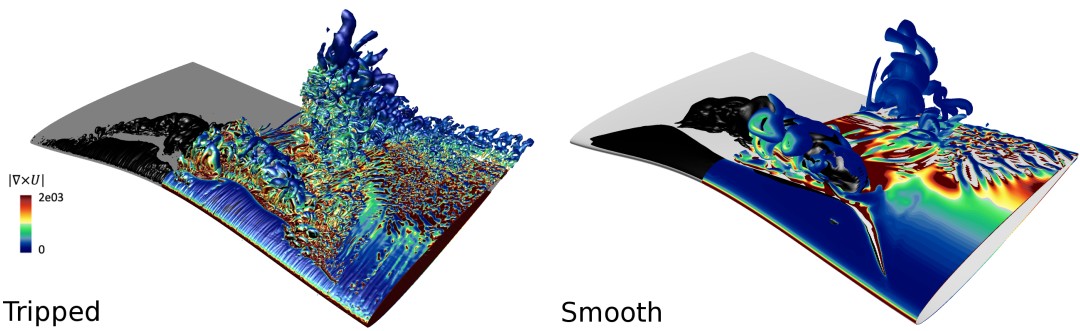

**Figure 8.** Isometric view of the numerical results of Figure 7e.

The pattern of the cloud cavity convection to the downstream is presented in Figure 9. In the first instance, Figure 9a, relatively similar sheet cavity and cloud cavity features are predicted in the smooth and tripped cases. However, along with the more accurate sheet cavity width, the cloud cavity shape agrees much better with the observations, especially on the side of the cloud.

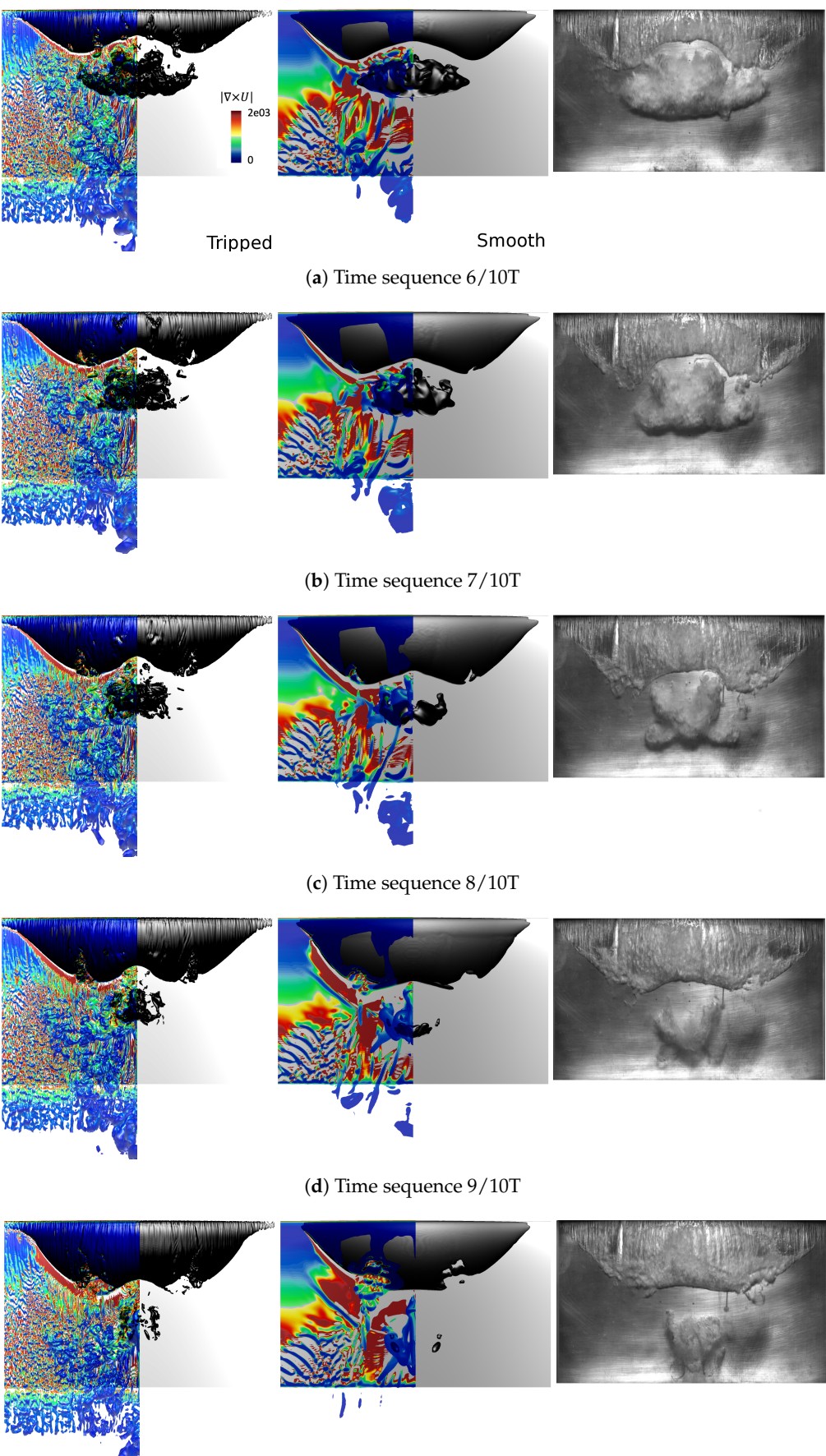

(**a**) Time sequence 6/10T

(**b**) Time sequence 7/10T

(**c**) Time sequence 8/10T

(**d**) Time sequence 9/10T

(**e**) Time sequence 10/10T

**Figure 9.** Cloud cavitation transportation and collapse.

When moving to the downstream and because of the pressure over prediction, the cloud cavity becomes smaller faster in the simulation compared to the experimental observation. The discrepancy of the cloud cavity size between the experimental data and the numerical results becomes more clear as the cloud reaches the trailing edge, Figure 9e. This simply implies that the condensation rate is much over-predicted in the numerical results, especially in the foil down stream region. There are several factors that contribute to the accuracy of condensation, such as condensation phase change model (modelling perspective), and computational time and spatial resolution (numerical perspective). The mass transfer model employed in the current study is derived from a single bubble dynamic without considering the interactions caused by the surrounding bubbles or cavities, while the cloud cavity dynamics involve strong iterations of collapsing bubbles. The later one includes bubble collapse jet, shock-waves and compressibility, which are not considered in the current approach.

As stated before, the separated cavity and the vortical structures generated behind the roughness elements contain very small interactive turbulent vortices; their pressure distribution prediction depends on the accuracy of the modelling, which itself strongly is subjected to the spatial mesh resolution. Lack of having a proper mesh resolution will result in the over-prediction of pressure, especially in the core of these vortices. This has higher impact on the tripped case results, where many small scale flow structures generated by roughness elements need to be accurately resolved to correctly convect the cloud cavity downstream.

Which one of the above-mentioned factors, i.e., the limitation in the condensation mass transfer model or the lack of having enough spatial resolution, is the main reason for the early collapse of the cloud cavity in the numerical results; this is not clear. For the bubbly cloud cavity predicted in the tripped case, as the cavitating structures are small and yet reasonably well represented, using a more advanced Lagrangian model, or more precisely, a hybrid Lagrangian–Eulerian cavitation model, can remove some of the drawbacks of the current mass transfer modelling [33].

A detailed example of vortical structures interacting with the cloud cavity is presented in Figure 10. The figure is an iso-metric view of the results related to the instance Figure 9c. It can be clearly noted that the cloud cavity contains many small bubbles clustered together to form a concentrated vorticity region, which is obviously not captured in the smooth case.

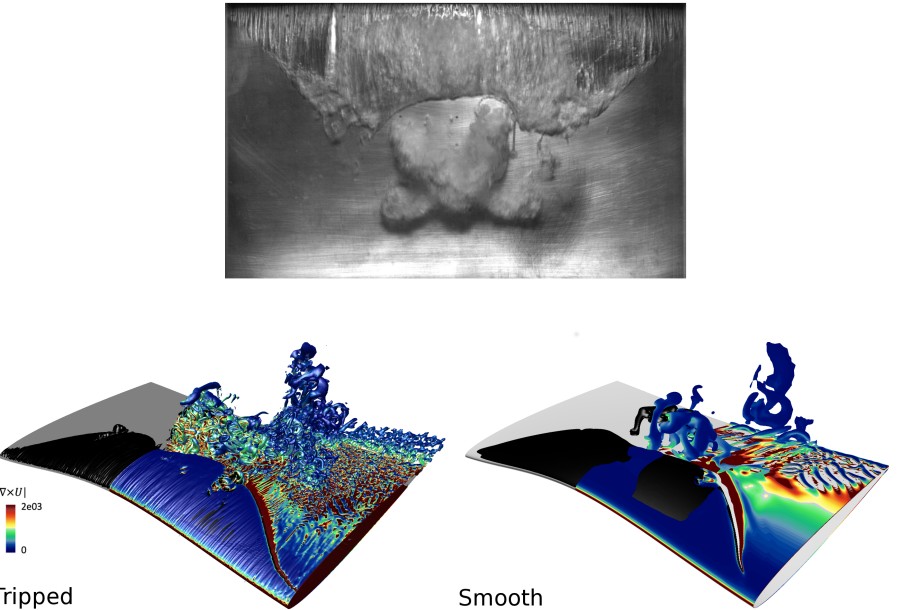

**Figure 10.** Isometric view of the numerical results of Figure 9c.

### 6.4. Cavitation-Vortex Interaction

In Figure 11, the average distribution of quantities related to the vorticity properties in the smooth and tripped cases is presented. The figure includes the distribution of magnitude of $Q$-criterion, vorticity, vortex stretching term, vortex dilatation term, and baroclinic term on the mid-section of the foil, $Y = 0.5$.

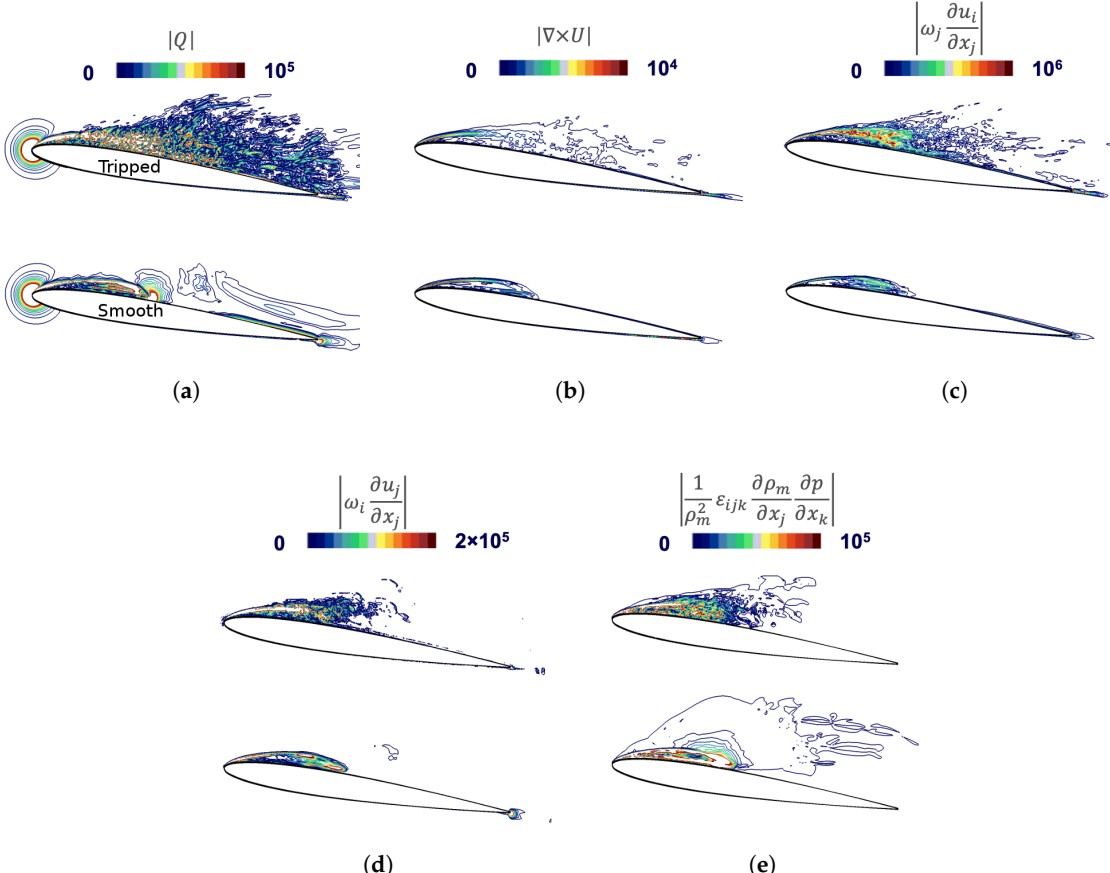

**Figure 11.** Distribution of vorticity properties for the smooth and tripped cases at Y = 0.5. (**a**) Magnitude of $Q$-criterion. (**b**) Magnitude of velocity curl. (**c**) Magnitude of stretching term. (**d**) Magnitude of dilatation term. (**e**) Magnitude of baroclinic term.

In Figure 11a, the $Q$-criterion absolute value is presented, where the presence of small structures all over the foil and especially in the region related to the cloud cavity formation is noticeable for the tripped case. It can be noted that the maximum $Q$-criterion value is observed to be as in the cavity closure region. Comparison of the vorticity magnitude, Figure 11b, indicates that the highest vorticity magnitude happens on the interface of the cavity, and it is larger and also sharper in the smooth case. This corresponds to having a less-distributed cavity interface in the smooth case. The transport of the vorticity, however, has been captured better in the tripped case where the vorticity field is extended to the trailing edge, representing the convection of the cloud cavity.

The vortex stretching term represents the enhancement of vorticity by stretching from one direction to another one, and therefore, it is essential to the 3-D structure of turbulence. The vortex stretching is the mechanism by which the turbulent energy is transferred to smaller scales. It also reduces the length scales of the turbulence in the two directions perpendicular to the turbulence, while intensifying the vorticity. The tripped case result of the vortex stretching, Figure 11c, clearly indicates the region where the sheet cavity cuts off and the cloud cavity forms is the main region for the energy transfer between different flow directions. It emphasises when the cloud forms the flow, which

becomes fully three-dimensional with a much smaller turbulence length scale, perpendicular to the free-stream direction.

In Figure 11d, the vorticity dilatation term is presented. As can be noted from its definition, its magnitude is related to the magnitude of the velocity divergence and the flow vorticity, and the vorticity dilatation consequently becomes noticeable in the regions where either of these quantities are significant. This term represents the effects of flow expansion on the vorticity field, and is non-zero when either the flow is compressible or mass transfer occurs. Where condensation happens, $\frac{\partial u_j}{\partial x_j}$ is a positive quantity and therefore this term would result in a decrease in the magnitude of vorticity due to the minus sign in front of this term. In the region where evaporation happens, this term acts as a positive source term, leading to an increase in the flow vorticity. It should be noted, however, that this is not a generation or destruction term in the sense of creating or destroying vorticity, but rather as an act to redistribute existing vorticity. For the studied foil, the evaporation is only active in a small region in the leading edge part and the rest of the domain experiences the condensation. Therefore, Figure 11d indicates the region where, due to the mass transfer, the vorticity has been redistributed among different flow directions.

The baroclinic torque, Figure 11e, represents the generation of vorticity as a result of non-aligned density and pressure gradients. For a flow with density gradient, if the pressure gradient is not parallel to the density gradient, the pressure field will lead to unequal acceleration over the variable density field. In a simpler word, the lighter density fluid will be accelerated faster than the high density fluid, resulting in a shear layer, and thus the generation of vorticity.

## 7. Conclusions

In this paper, effects of the leading-edge roughness on the flow vortices development and cloud cavitation formation are investigated by the numerical modelling of turbulent cavitating flows around the Delft Twist11 foil. Because of the spanwise varying angle of attack, the natural transition of the laminar to turbulence boundary layer and consequently the cavitation inception depend on the spanwise direction. In the experimental tests conducted in the Delft Technical University, and used as the benchmark of Workshop on Cavitation and Propeller Performance at the Second International Symposium on Marine Propulsors, leading-edge roughness is applied to effectively force the transition to turbulence along with generation of enough flow nuclei. This has previously not been considered in numerical simulations of the flow. The roughness provides a more complex three-dimensional mechanism of cavitation shedding, which itself depends on the vortical flow generated by the roughness elements. This flow development has previously been hypothesised based on the experiments, and is here corroborated by the simulation.

The turbulence is modelled by using an incompressible Implicit large eddy simulation, along with the homogeneous assumption for the two-phase mixture flow. The employed numerical settings and mesh resolution follow the guideline concluded from our previous studies of this foil. The leading-edge roughness is geometrically represented in the computational mesh by a randomised removal of cells in the region, where sand roughness was applied in the experiment. We can note that this procedure gives a good qualitative representation of the effect of the roughness on the flow field.

Our results indicate a strong dependency of the cloud cavitation on the predicted vortical structures, especially at the region where the sheet cavity cuts off and the cloud cavity forms. In the case of the smooth foil, where the interaction of the leading edge vortical structures and the sheet cavity are very small, the cut-off of the sheet cavity forms a coherent separated bucket-shaped cavity, rather than a cloud-like cavity. Furthermore, the shape and richness of spatial scales are in better agreement with the experiments when using roughness, as well as the span-wise extent of the attached sheet.

The differences in the shedding process are found to have a large impact not only on the cavitation pattern but also on the pressure distribution. In the tripped case, the cloud cavity consists of several small-scale structures that, as the cavity travels downstream, gradually collapse. Since their cavity size

is small, the resulting effect on the pressure field is small compared to the main cloud cavity pressure field leading to a gradually constant increase in the pressure over the foil surface.

It is clear from the presented simulation that it is important to mimic the experimental set-up closely to reduce the modelling errors and improve the interpretation of simulation, as well as experimental, results.

**Author Contributions:** R.E.B. and A.A. conceived of the presented idea, developed the theoretical framework, developed the analysis tool, and performed the analysis; A.A. made the simulations, collected the data and outlined the paper; R.E.B. was involved in planning and supervised the work; R.E.B. and A.A. contributed to the analysis of the results and to the writing of the manuscript; R.E.B. was responsible for providing the computing resources, project administration and funding acquisition. All authors have read and agreed to the published version of the manuscript.

**Funding:** This research received no external funding. However, financial support for this work has been provided by Kongsberg Maritime Sweden AB through the University Technology Centre in Computational Hydrodynamics hosted at the Department of Mechanics and Maritime Sciences at Chalmers.

**Acknowledgments:** The simulations are performed on resources at Chalmers Centre for Computational Science and Engineering (C3SE) provided by the Swedish National Infrastructure for Computing (SNIC).

**Conflicts of Interest:** The authors declare that there is no conflict of interest.

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
