# Peer review of "Impact of Leading Edge Roughness in Cavitation Simulations around a Twisted Foil"

_fluids, doi:10.3390/fluids5040243_

Round 1
Reviewer 1 Report
In the current article the authors present a computational analysis of a cavitating flow over the Delft twisted foil in the smooth as well as the rough regime.
The numerical results have been compared with experiments.
The authors have performed commendable work for this investigation. This article will add substantial knowledge to the community. There are just a few minor comments.
The abstract is well written and clearly summarizes the paper. The introduction too places the study very well in context with the existing work clearly outlining the objective of the research.
The introduction however points to a previous work done by the authors which establishes the numerical set up of the simulation. The authors are requested to provide a short summary of the findings to make it easier for the reader to follow.
Also, the study does not mention any assumptions made. The authors are requested to kindly specify the assumptions here.
Figure 7 does not show a vorticity scale. Please add a scale to distinguish the magnitude.
The conclusions support the aims and findings of the study and the work overall is impressive.
Reviewer 2 Report
The paper by Asnaghi and Bensow deals with the cavitating flow induced by roughness elements on the leading edge of a twisted foil, in the framework of fully-developed three-dimensional turbulence. By means of theoretical calculations and mainly numerical computations (identical to the experimental tests performed at Delft TU and similar to those of EPFL), the roles of the different parameters into play are investigated. After recalling the analytical modelling behind implicit large-eddy simulations and phase change, and describing the empirical geometry and the cell removal procedure, the authors introduce the OpenFOAM libraries with PIMPLE algorithm, and study the pressure and lift coefficients and the shedding\collapse pattern of the bubble cloud. Focusing then on the influence of such structures on the vorticity field, they show the spatial distribution of the so-called "Q-criterion", and assess the magnitude of the three main contributions - stretching, dilatation, baroclinicity - in the transport equation. Their results are carefully presented and plotted, and the relevance of these findings with respect to the existing bibliography from scientific literature is thoroughly discussed.
In my opinion, the article deserves publication on Fluids. Hereafter I only mention a few minor issues and typos, to be clarified or amended by the authors.
1) Line 1: "Twisted" -> "twisted"
2) Lines 26-27: singular\plural mismatch in "a ... tests"
3) Lines 55-56: singular\plural mismatch in "A detailed comparison ... are conducted"
4) Line 59: "an" -> "a"
5) Beginning of section 2: here and in the rest of the paper, please use the hyphen consistently in all constructions of the type ADJECTIVE-NOUN NOUN ("single-fluid ... mixture")
6) In equation (2) please clarify the role of the gravity force and the absence of the buoyancy response: is the latter neglected, or is Archimedes' contribution included in some stress tensor, or is everything taking place in vacuum?
7) In equation (7) and in line 68, please specify the meaning of the quantities $p_{threshold}$, $\alpha_{Nuc}$, $R_B$, $L_{\infty}$, $U_{\infty}$.
8) In equation (8), the quantity inside vertical bars is a matrix (not a vector); if extracting its norm is the operation meant here, please use double vertical bars.
9) End of subsection 2.1: please specify the influence of the quantities $n_0$ and $d_{Nuc}$ on the terms of the previous equations.
10) Line after equation (9): "Alternating Tensor" -> "alternating tensor"
11) Line 80: here and in the rest of the paper, please use slanted\italic - not upright\roman - fonts for mathematical quantities (e.g. here $Q$)
12) Line 100: "and the" -> "the"
13) Lines 104-105: singular\plural mismatch in "streaks ... elements ... affects"
14) Line 130: "an" -> "a"
15) Line 136: here and in the rest of the paper, please use upright\roman - not slanted\italic - fonts for text acronyms (e.g. here RR1)
16) In the label of figure 3 and in line 142, please specify the meaning of the quantity $S$ (which I suppose is not related to the strain rate tensor)
17) Lines 162-163: singular\plural mismatch in "the number ... are restricted"
18) Line 164: singular\plural mismatch in "their geometry are limited"
19) In line 187 please clarify the definition of "pressure coefficient"
20) In line 203 please clarify what is exactly meant by "higher", since such an adjective can be misunderstood when applied to a negative quantity such as $C_P$.
21) Line 269: "gradually" -> "gradual"
22) Lines 271-272: singular\plural mismatch in "stored energy ... are assumed"
23) Lines 303-304: singular\plural mismatch in "average distribution ... are presented"
24) Line 354: "based the" -> "based on the"
25) Line 367: singular\plural mismatch in "The differences ... is found"
